# Early experience affects foraging behavior of wild fruit bats more than their original behavioral predispositions

Adi Rachum[1], Lee M Harten[1], Reut Assa[1], Aya Goldshtein[1†], Xing Chen[1], Nesim Gonceer[1], Yossi Yovel[1,2*]

[1]School of Zoology, Faculty of Life Sciences, Tel Aviv University, Tel Aviv, Israel; [2]Sagol School of Neuroscience, Tel Aviv University, Tel Aviv, Israel

## eLife Assessment

This paper provides **important** insight into how early life experience shapes adult behavior in fruit bats. The authors raised juvenile bats either in an impoverished or enriched environment and studied their foraging behaviors. The evidence is **convincing** that bats raised in enriched environments are more active, bold, and exploratory. The work will be of interest to ethologists and developmental psychologists.

**\*For correspondence:**
yossiyovel@gmail.com

**Present address:** †Max Planck Institute of Animal Behavior, Department of Collective Behavior, Konstanz, Germany

**Competing interest:** The authors declare that no competing interests exist.

**Abstract** There are immense consistent inter-individual differences in animal behavior. While many studies have documented such behavioral differences, often referred to as individual personalities, little research has focused on the underlying causes and on determining whether they are innate or based on individual experience. Moreover, most studies on animal personalities have described consistent differences in behavior under laboratory conditions. We aimed to examine the impact of the early experienced environment on individual animal behavior, and to compare it to that of the individual's original genetic predisposition. Additionally, we explored the correlation between personality traits measured indoors and the animal's outdoor behavior. We studied Egyptian fruit bats, in which vast behavioral variability and plasticity have already been demonstrated. We raised bats in a captive colony under either enriched or impoverished environments and assessed their personality under controlled laboratory conditions. We then released the bats into the wild and tracked their foraging using GPS. Bats that had experienced an enriched environment during early life displayed increasing boldness and exploratory behavior when foraging outdoors, demonstrating how early-life experience can affect adult behavior. The individuals' original predispositions did not predict their later foraging behavior. Our findings shed new light on the interplay between innate and experience-based effects on individual behavior.

## Introduction

Anyone who has ever observed animals knows that individuals persistently differ in their behavior. These behavioral tendencies, enable distinguishing individuals from one another and remain consistent over time and across various situations, have been termed 'animal personalities' (*Stamps and Groothuis, 2010*). While some opposition exists regarding the use of this term (*Beekman and Jordan, 2017*), it is nonetheless commonly employed (*Hertel et al., 2020*; *Roche et al., 2016*), and we use it here to describe a set of behavioral traits that remain consistent within individuals over time and across contexts (*Stamps and Groothuis, 2010*).

Personality traits have been extensively documented in various taxa (*Stamps and Groothuis, 2010*; *Kazlauckas et al., 2005*; *Menzies et al., 2013*). They are typically assessed through the measurement of proxies for behavioral traits such as boldness, activity, and exploration (*Réale et al., 2007*). However, while individual personality differences have been observed in numerous species (*Kazlauckas et al., 2005*; *Menzies et al., 2013*; *Freeman and Gosling, 2010*; *Speechley et al., 2024*), only a limited number of studies have investigated the underlying factors that contribute to their emergence.

Broadly speaking, the ontogeny of an animal's personality can be affected by its genetic predisposition and by its individual experience. To date, only a few studies have explored how individual personality develops, and how the interplay between genetic factors, maternal effects, and early experiences or environmental effects shape personality (*Stamps and Groothuis, 2010*; *Menzies et al., 2013*; *Harten et al., 2021*; *Guenther et al., 2014*; *Xu et al., 2021*). In the few studies that did so, the effects of these factors have been observed exclusively in captive settings (*Xu et al., 2021*; *Crawford et al., 2020*; *Díaz-Fleischer et al., 2009*; *White and Brown, 2015*; *Zaias et al., 2008*). For instance, researchers found differences in personality traits, such as boldness and exploration, in guinea pigs raised in captivity under different photoperiods simulating spring or autumn conditions (*Guenther et al., 2014*). Similarly, intertidal gobies reared in different habitat complexities have shown differences in spatial learning laboratory experiments: gobies that developed in complex rock pools demonstrated faster learning abilities compared to those raised in homogenous sandy shores (*White and Brown, 2015*).

Only a handful of studies have examined behavioral traits under both captive and natural conditions. One such experiment showed that the foraging behavior of birds in captivity was consistent with their behavior in the field. For example, the exploratory tendencies in captivity were correlated with seeking new feeding sites in the wild (*Herborn et al., 2010*). There is also a scarcity of hypothesis-driven experiments that employ specific manipulations to investigate the ontogeny of personality. In this study, we aimed to elucidate the roles of genetic predisposition and early life experience on free-foraging fruit-bat personality using a controlled manipulation.

Egyptian fruit bats have been shown to exhibit immense individual differences both in indoor personality tests (*Harten et al., 2018*) and when foraging in the wild (*Harten et al., 2020*). These bats have also been shown to exhibit environmentally dependent behavioral plasticity. Specifically, the same individual fruit bats were much more exploratory when foraging in urban environments than when foraging in rural environments (*Harten et al., 2021*; *Egert-Berg et al., 2021*). Moreover, fruit bats that roost in cities were shown to be even more exploratory than rural fruit bats that commute into cities, suggesting that the environment might have an additive effect on their behavior.

In this study, we performed a controlled manipulation to determine whether the environment that young bat pups experience can affect their personality and outdoors foraging. Specifically, we examined whether exposure to an enriched environment as pups affects the bats' behavior as measured in the lab and when they later forage in the wild as adults. Living in enriched or dynamically changing environments has been linked in multiple species to many behaviors, including increased risk-taking and boldness (*Harten et al., 2021*; *Fehlmann et al., 2017*; *Weimerskirch et al., 2023*). Maturation in an enriched environment can also affect learning capabilities, as noted above regarding gobies (*White and Brown, 2015*), and motor performance as demonstrated in laboratory rats (*Crawford et al., 2020*).

In our study, we manipulated the extent of environmental enrichment experienced by juvenile fruit bats in a captive environment and examined the correlation between this early experience and their foraging behavior in the wild several months later. We compared the effects of experience to the bats' original behavioral predisposition, estimated when they were naïve pups (~4 months old; prior to any manipulation). We also examined personality traits over a prolonged period to confirm their individual behavioral consistency, and we explored the relationship between the personality traits observed in captivity and the outdoor foraging behavior. We hypothesized that both early-life experience and original predisposition would interact in shaping individual personalities.

## Results

### Assessing personality in the lab

To assess bats' personality, we ran young bats through the multiple-foraging box paradigm, which was developed in our lab and has been shown to measure behavioral variability along multiple axes (*Harten et al., 2021*). A total of 40 bats participated in the foraging box experiments, which were repeated a maximum of five times per individual over a period of almost 5 months (*Figure 1A*, see Methods). The foraging box experimental setup was designed to assess the innate personality traits in bats, which we could then use as parameters when trying to explain their foraging outdoors. Repeated measurements were used to confirm that the traits met the criteria for personality, such as consistency over time. The experiment comprised baseline trials for each bat on two consecutive nights, upon the bats reaching an average age of 4.1±1.4 months (trials 1–2; Mean ± STD). Subsequently, we randomly divided the bats into two colonies, each exposed to different environmental conditions. Both colonies were provided with the same basic diet sufficient for all bats in the colony. Importantly, bats in both colonies were raised in large social groups of similar size. The *enriched colony* underwent frequent changes in their surroundings for a period of ~10 weeks, through the introduction of various enrichments that forced the bats to practice trial and error to obtain (Methods). While the *control impoverished colony* was exposed to a stable environment with minimal changes. A post-enrichment boxes trial was performed immediately after this period (Trial 3, on average 72 days after the previous). Finally, two post-release box trials were performed after the bats were released into our open colony and could freely forage in the wild and explore their natural environment (Trials 4–5, average 45 and 58 days after their release into the wild). Post-enrichment and post-release trials were conducted to assess whether early life experiences had an effect on the development of bat personality and in order to identify persistent personality traits that can be used as a proxy of the individual innate predispositions. We used these trials to assess proxies of the three above-noted behavioral traits: boldness, exploration, and activity. To assess the behavioral traits during the foraging box experiments, we scored the bats' actions. Specifically, we recorded how many times they approached and took food from a foraging box, the number of different boxes they explored (i.e. landed on), and the total number of actions they performed during the night (*Figure 1A* and see Methods).

When plotting the three behavioral traits (boldness, exploration, and activity) in 3D, they seem to form a triangle (*Figure 1B*) – reminiscent of a Pareto front (*Shoval et al., 2012*) – suggesting a trade-off between the different traits. The three vertices of the triangle (often referred to as the archetypes when analyzing a Pareto front) represent individuals that exhibit: (a) high exploration, high activity, and medium-low boldness; (b) high boldness, low activity, and low exploration; and (c) low boldness, low activity, and low exploration (very few bats). These three archetypes suggest a trade-off between boldness on the one hand, and activity and exploration on the other hand.

For all three behavioral traits, we found a significant positive correlation between baseline (Trial 1) and the post-enrichment trial (Trial 3) examined on average 72 days apart. A significant positive correlation was also found between bats' boldness during the baseline (Trial 1) and the second post-release trial (Trial 5), examined on average 144 days apart. These findings suggest that these behavioral traits, and especially boldness, remain consistent at least throughout the bat's early life (*Figure 1C1-3*, Pearson correlation test: $r=0.66$ $p<9.7e-0.5$, $r=0.66$ $p<9.1e-0.5$, and $r=0.51$ $p<0.004$; for Trials 1–3 for boldness, exploration, and activity, respectively, and $r=0.6$ $p<0.021$, for Trials 1–5 boldness, *Table 1*). These results did not change when examining the enriched and impoverished colonies separately. To compute an integrative measurement of personality, we ran a PCA analysis on the data from trials 1 and 5. We found a significant Pearson correlation ($p=0.03$, $r=0.58$, *Figure 1—figure supplement 1*) between the individuals' scores on PC1 in trials 1 and 5 suggesting that this measure captures a persistent representation of behavior that takes both boldness and exploration into account (the activity weight was very low).

Notably, all three personality traits exhibited significant correlations in the two baseline trials suggesting that indeed they capture behavioral tendencies (*Figure 2*).

The temporal dynamics of boldness and exploration revealed similar patterns: both increased between the first two consecutive baseline trials (*Figure 2*). Following the enrichment condition (i.e. in Trial 3), they decreased back to the level of the first baseline and then increased again in the two post-release trials (4-5). Boldness was significantly higher in the last trial compared to the first one ($p<1.9e-07$, see discussion). A mixed-effect Generalized Linear Model (GLMM) was used, with the

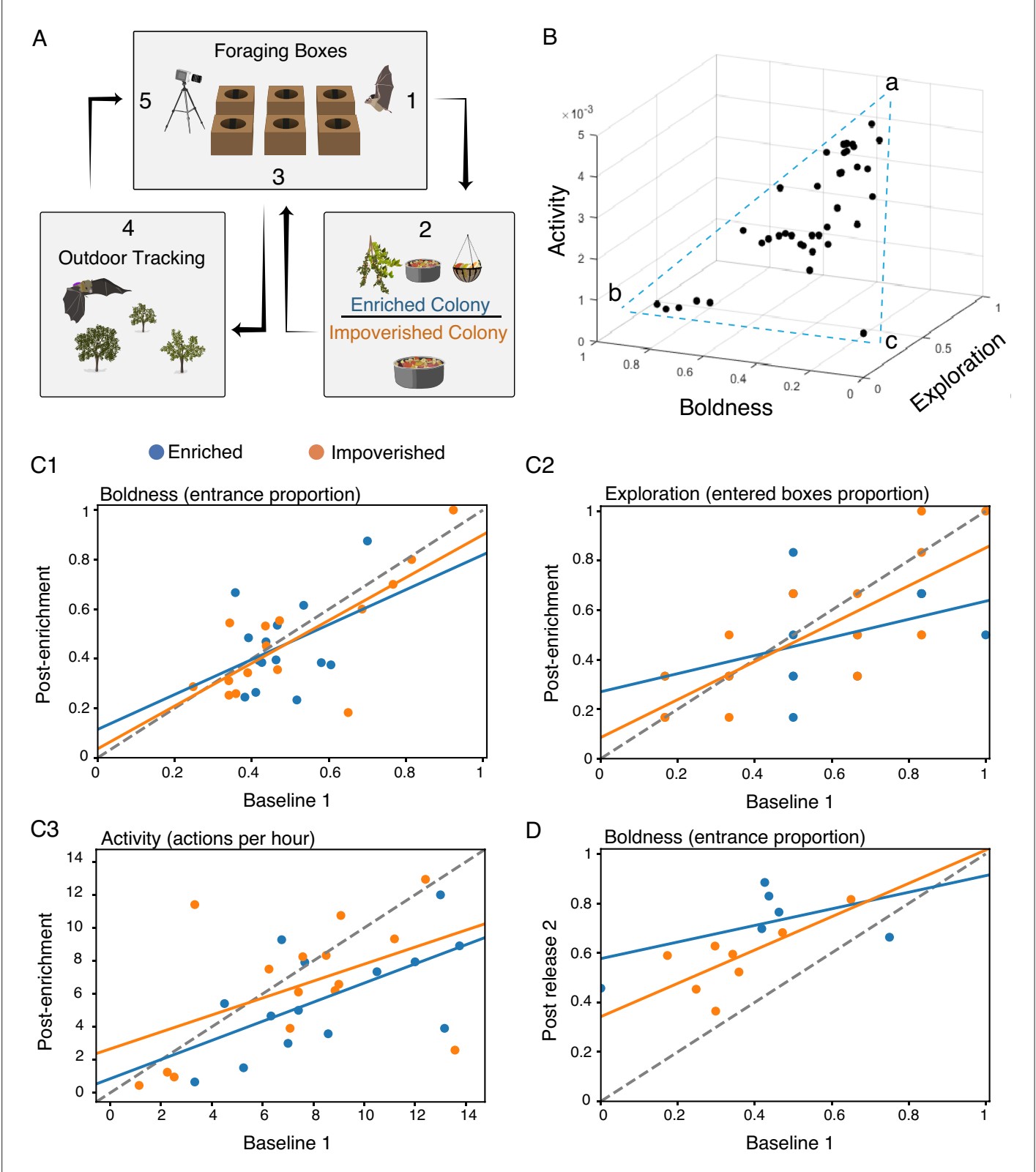

**Figure 1.** Assessing personality in the laboratory. (**A**) Schematic illustration of the experiments. (**B**) A 3D plot of the three behavioral traits, estimated during Trial 1. (**C1-3**) Behavioral traits were consistent over time. The behavioral traits of the first and third (post-enrichment) trial are presented. Positive correlations were found for all three behavioral traits over a period of more than 10 weeks. For the enriched (blue lines), the impoverished (orange lines),

*Figure 1 continued on next page*

*Figure 1 continued*

and both environments together (not shown), respectively: (**C1**) Boldness; (**C2**) exploration; and (**C3**) activity. (**D**) Boldness of the first and fifth (post-release 2) trial. Dashed line represents the Y=X line.

The online version of this article includes the following figure supplement(s) for figure 1:

**Figure supplement 1.** PCA analysis for trial 1 (baseline) and 5 (post-release 2); positive correlation was found between the individual scores on PC1 (Pearson correlation *p*=0.03, *r*=0.58).

behavioral traits set as the response variable, trial number (1 or 5) as a fixed factor, and bat ID as a random effect (see *Table 2*). While exploration showed a fairly similar pattern. Activity, in contrast, showed a different pattern. It decreased continuously throughout the trials and was significantly lower in the last trial compared to in the first one (*p*<0.0001, GLMM as above). An individual-level comparison between the first and last trial is presented in *Figure 2C*.

Despite the above-reported changes in behavior over time, the environment that the animals experienced as juveniles (enriched or impoverished) did not have a significant effect on their personality as measured in the lab (GLMM - with the behavioral traits set as the response parameter, the trial number and the interaction between trial number and the environmental condition as fixed factors, and bat ID as a random effect). The interaction between trial and environmental condition was not significant (*Table 3*). When examining only Trials 4–5, i.e., following their release into nature, the enriched bats were observed to be significantly bolder than the non-enriched bats (*p*=0.003, GLMM as above). However, this seems like a result of post-selection, resulting from which bats remained in our colony and could be tested (see Discussion).

## Outdoor foraging behavior

After releasing the bats into our open colony, in which they were free to fly out, we tracked them using GPS devices as they explored the world for the first time and foraged in their natural environment. We analyzed data from 19 bats, obtaining data from 39.2±17.0 (Mean ± SD) foraging nights per bat on average, accounting for 72.5±8.3% (Mean ± SD) of the individuals' foraging nights. We used the following movement parameters as representatives of the bats' outdoor behavior: the total nightly time spent foraging as a proxy for activity; the explored area as a proxy for exploration; and the maximal distance from the colony as a proxy for boldness (see Methods).

The enriched bats significantly differed from the impoverished bats in all these movement parameters (*Figure 3*, *Figure 3—figure supplements 1 and 2*). Specifically, they spent significantly more time out foraging every night (3.5±1.9 vs 2.8±1.8 hr), flew farther from the colony (1.3±1.8 vs 0.8±1.13 km), and explored larger areas (7.82±6.72 vs 3.39±3.86 km$^2$); all values represent the Mean ± SD for the entire study period (19 bats). In this analysis, we included all bats and averaged according to the number of nights we recorded per bat. Examining the data after 15–20 nights outside, where all bats have been out for a similar period (n=17 bats with data), the pattern was the same and the values were: 4.02±1.19 vs 2.8±0.75 hr, 1.66±1.77 vs 0.6±0.28 km, 2.34±2.71 vs 0.23±0.23 km$^2$.

To assess the relative effects of experience (i.e. enriched/impoverished) on foraging and to compare it to the bats' original predisposition, we ran a GLMM, with their original predisposition (Trial 1) and environment (enriched/impoverished) as fixed factors. We ran this analysis two times using either boldness or the PC1 score that accounted for both boldness and exploration of Trial 1 as a proxy for the bats' innate predisposition, because these had been examined before the bats were exposed to the different environments.

The analysis revealed that the environment in which the bats were reared was more important than their original predisposition for explaining their outdoor foraging behavior. While the effect of predisposition was not significant, the enriched environment significantly correlated with spending more time outside, flying to a significantly larger maximal foraging distance, and exploring a larger

**Table 1.** Pearson's correlation test.

|  | **Boldness 1–3** | **Exploration 1–3** | **Activity 1–3** | **Boldness 1–5** |
|---|---|---|---|---|
| Pearson r value | 0.66 | 0.66 | 0.51 | 0.6 |
| Pearson p-value | <9.7e-05 | 9.1e-0.5 | 0.004 | 0.021 |

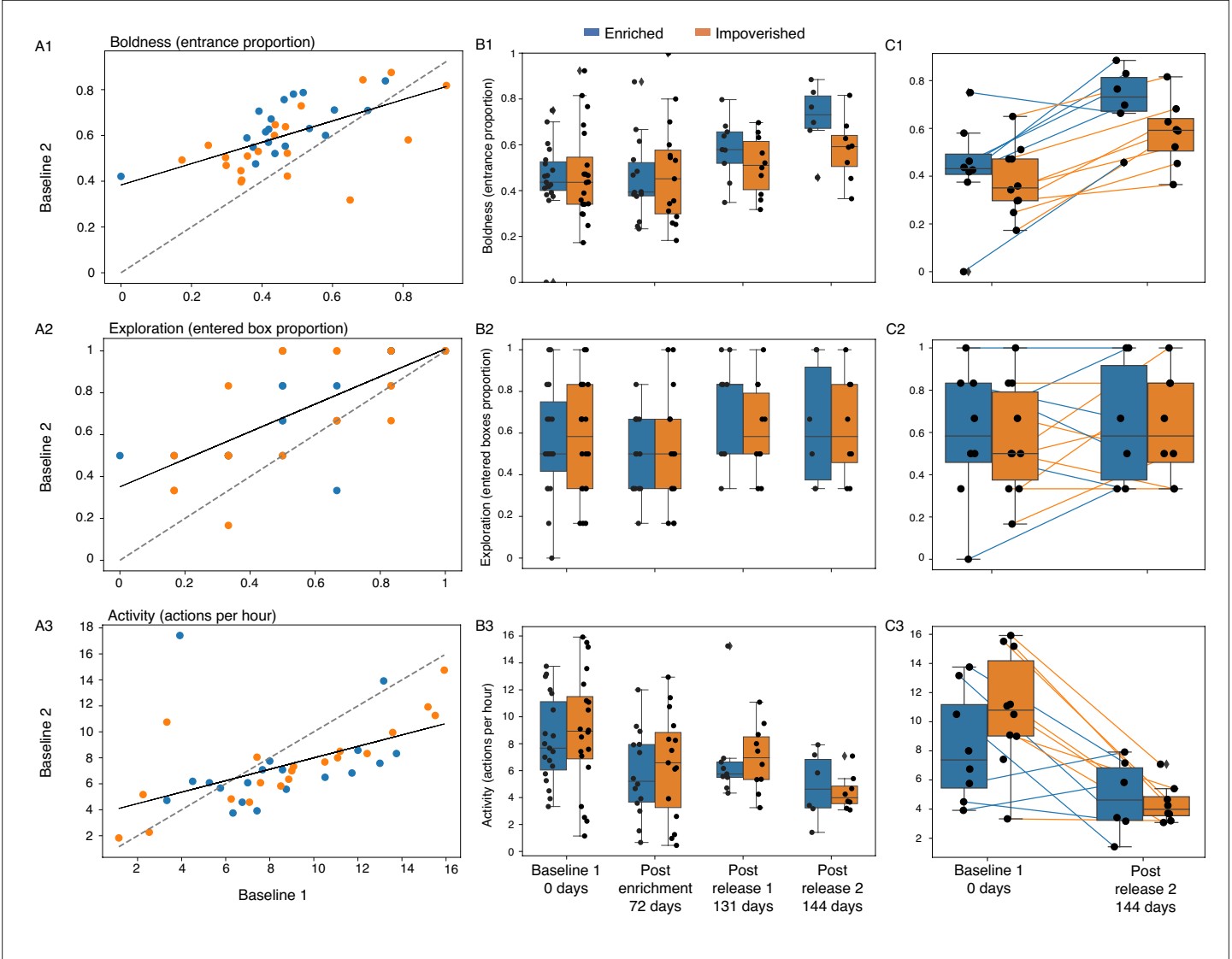

**Figure 2.** Personality measurements over time. (**A**) The behavioral traits of the first and second baseline trials, performed on two consecutive nights, are presented. Positive correlations were found for all three behavioral. Boldness (Pearson *r*=0.59, *p*=5.8e-05) (**A1**); exploration (Pearson *r*=0.70, *p*=5.3e-07) (**A2**); and activity (Pearson *r*=0.52, *p*=0.0005) (**A3**). Dashed line represents the Y=X line. (**B**) Box plots depicting the bats' behavior across the five trials according to three traits: Boldness (**B1**), exploration (**B2**), and activity levels (**B3**). (**C**) Paired box plots show individual bat values in the first trial (baseline 1) and the fifth trial (post-release 144 days apart on average) for each trait: Boldness (**C1**), exploration (**C2**), and activity levels (**C3**). Lines connect repeated measurements from the same individual, illustrating within-individual behavioral changes over time. The number of bats participated in each trial is given in Methods, *Table 8*. In the baseline trials, no significant differences were found between the two environmental conditions (enriched / impoverished) for any of the three behavioral traits we measured (*p*=0.62, *p*=0.84, *p*=0.72 for boldness, exploration, and activity levels, respectively, n=40, *Supplementary file 1*). *Figure 2—figure supplement 1* also shows the indication. The experimental phase and the time in days from the day of the first trial are depicted on the X-axis. Box plots show the 25% and 75% percentiles. Medians and whiskers based on 1.5 IQR are shown.

The online version of this article includes the following figure supplement(s) for figure 2:

**Figure supplement 1.** Full bat behavioral traits over time.

area (*p*<0.028, *p*<0.01, and *p*<0.02[1], respectively). GLMM was used (see *Tables 4 and 5*, with environmental condition as the predisposition and experience as fixed factors). A model that includes an interaction between the predisposed boldness, exploration, or activity and the environmental condition showed a worse fit (higher AIC). The bats' experience (represented by the number of nights they spent foraging until the day of the assessment) also had a significant positive effect, increasing the nightly foraging time and the maximal distance (*p*<0.0006 and *p*<0.004; GLMM for the time spent

**Table 2.** Mixed generalized linear model (GLM) results for the comparison between trials 1 and 5 response ~trial + (1 | bat ID).

| Response | AIC | BIC | LogLikelihood | Deviance | | | |
|---|---|---|---|---|---|---|---|
| | –36.379 | –28.498 | 22.189 | –44.379 | | | |
| | Fixed effects coefficients (95% CIs) | | | | | | |
| | Name | Estimate | SE | tStat | DF | p-value | Lower | Upper |
| | Intercept | 0.408 | 0.031 | 13.16 | 51 | 5.03e-18 | 0.346 | 0.471 |
| Boldness | Trial number | 0.056 | 0.009 | 6.02 | 51 | 1.902e-07 | 0.037 | 0.075 |
| Response | AIC | BIC | LogLikelihood | Deviance | | | |
| | 12.074 | 19.956 | –2.037 | 4.074 | | | |
| | Fixed effects coefficients (95% CIs) | | | | | | |
| | Name | Estimate | SE | tStat | DF | p-value | Lower | Upper |
| | Intercept | 0.545 | 0.049 | 11.02 | 51 | 4.248e-15 | 0.446 | 0.644 |
| Exploration | Trial number | 0.018 | 0.016 | 1.157 | 51 | 0.252 | –0.013 | –0.013 |
| Response | AIC | BIC | LogLikelihood | Deviance | | | |
| | –580.34 | –572.46 | 294.17 | –588.34 | | | |
| | Fixed effects coefficients (95% CIs) | | | | | | |
| | Name | Estimate | SE | tStat | DF | p-value | Lower | Upper |
| | Intercept | 0.002 | 0.0001 | 13.737 | 51 | 9.050e-19 | 0.002 | 0.003 |
| Activity | Trial number | –0.0002 | 7.135e-05 | –4.065 | 51 | 0.0001 | –0.0004 | –0.0001 |

outside and the maximal nightly distance, respectively, *Tables 4 and 5*). The bats' age (days from birth) did not have a significant effect. However, the proportion of nights that bats spent foraging outside the roost was not significantly associated with environmental condition, nor was it influenced by sex or age (*p*=0.95, see *Supplementary file 2*).

We also tested the correlation between the behavioral traits measured in the wild and found a significant positive correlation only between the maximal distance and the area the bat explored, (Pearson correlation test; *Table 6*).

Lastly, we compared the foraging behavior outdoors to the behavioral personality traits measured in the indoor foraging task (assessed ~ 2 weeks following their release into the open colony - Trial 4). There was no significant correlation between indoor personality and any of the outdoor foraging characteristics (Pearson correlation test, *p*>0.05 for all comparisons).

## Discussion

Intra-specific inter-individual behavioral differences have been documented in many species (*Kazlauckas et al., 2005*; *Menzies et al., 2013*; *Freeman and Gosling, 2010*), but very few studies have explored the processes leading to these differences. In this study, we used a controlled manipulation to examine the effects of the early-life environment to which juvenile fruit bats had been exposed on their adult behavior.

We demonstrate that early exposure to an enriched environment seems to affect the foraging behavior of bats in the wild. Bats that were exposed early on to an enriched environment were bolder (i.e. flew farther from home) and were more exploratory. Preliminary results suggest that these differences may persist throughout the bat's lifetime; however, additional data must be analyzed to determine the long-term effects. Moreover, their early exposure was a better predictor of their foraging behavior than their individual behavioral predisposition that had been measured indoors before they had any experience of the outside world. We assessed the bats' predispositions at a very young age, as soon as they could fly but before they had a chance to gain any experience. These dispositions are thus likely innate, but we note that they might also have been shaped by maternal effects (e.g. by hormonal transfer, see *Harten et al., 2021*) in addition to genetics. Importantly, all personality

**Table 3.** Mixed generalized linear model (GLM) results for all tested trials (1-5) response ~1 + Trial × EnvironmentalCondition + (1 | bat ID).

| Response | AIC | BIC | LogLikelihood | Deviance | | | | |
|---|---|---|---|---|---|---|---|---|
| Boldness | −100.5 | −82.807 | 56.25 | −112.5 | | | | |
| | Fixed effects coefficients (95% CIs) | | | | | | | |
| | Name | Estimate | SE | tStat | DF | p-value | Lower | Upper |
| | Intercept | 0.448 | 0.0444 | 10.078 | 137 | 3.266e-1 | 0.36 | 0.536 |
| | Environmental condition Impoverished | 0.007 | 0.061 | 0.117 | 137 | 0.906 | −0.115 | 0.129 |
| | Trial number | 0.038 | 0.014 | 2.744 | 137 | 0.006 | 0.010 | 0.066 |
| | Environmental condition Impoverished: Trial | −0.011 | 0.019 | −0.58 | 137 | 0.561 | −0.049 | 0.026 |
| Response | AIC | BIC | LogLikelihood | Deviance | | | | |
| Exploration | −0.46128 | 17.231 | 6.230 | −12.461 | | | | |
| | Fixed effects coefficients (95% CIs) | | | | | | | |
| | Name | Estimate | SE | tStat | DF | p-value | Lower | Upper |
| | Intercept | 0.625 | 0.066 | 9.467 | 137 | 1.13e-16 | 0.494 | 0.755 |
| | Environmental condition Impoverished | −0.010 | 0.092 | −0.112 | 137 | 0.910 | −0.192 | 0.171 |
| | Trial number | −0.003 | 0.019 | −0.182 | 137 | 0.855 | −0.041 | 0.034 |
| | Environmental condition Impoverished:Trial | 0.0030.006 | 0.025 | 0.247 | 137 | 0.804 | −0.043 | 0.054 |
| Response | AIC | BIC | LogLikelihood | Deviance | | | | |
| Activity | −1586 | −1568.3 | 799.01 | −1598 | | | | |
| | Fixed effects coefficients (95% CIs) | | | | | | | |
| | Name | Estimate | SE | tStat | DF | p-value | Lower | Upper |
| | Intercept | 0.002 | 0.0002 | 11.017 | 137 | 1.319e-2 | 0.002 | 0.003 |
| | Environmental condition Impoverished | 0.0002 | 0.0003 | 0.908 | 137 | 0.365 | −30e-5 | 0.0009 |
| | Trial number | −0.0002 | 7.292e-05 | −3.273 | 97 | 0.001 | −30e-3 | −9.e-05 |
| | Environmental condition Impoverished:Trial | −9.2e-05 | 9.640e-05 | −0.959 | 137 | 0.338 | −20e-5 | 9.8e-05 |

traits exhibited significant correlations in the two baseline trials, suggesting that indeed they capture behavioral tendencies whose values might change with experience, while individual relations remain similar over time. Moreover, it is possible that a different set of personality traits would have predicted outdoor behavior better than those we used here, although we should note that we chose personality traits commonly used in such studies.

Our findings are in line with the findings from previous research of other organisms that showed that individuals raised in enriched environments (*Xu et al., 2021*; *Crawford et al., 2020*; *White and Brown, 2015*; *Zaias et al., 2008*) exhibited higher levels of boldness and exploratory behavior. Moreover, studies have shown that exposure to enriched environments enhances motor skills in animals (*Crawford et al., 2020*) and growing up in complex environments has been associated with

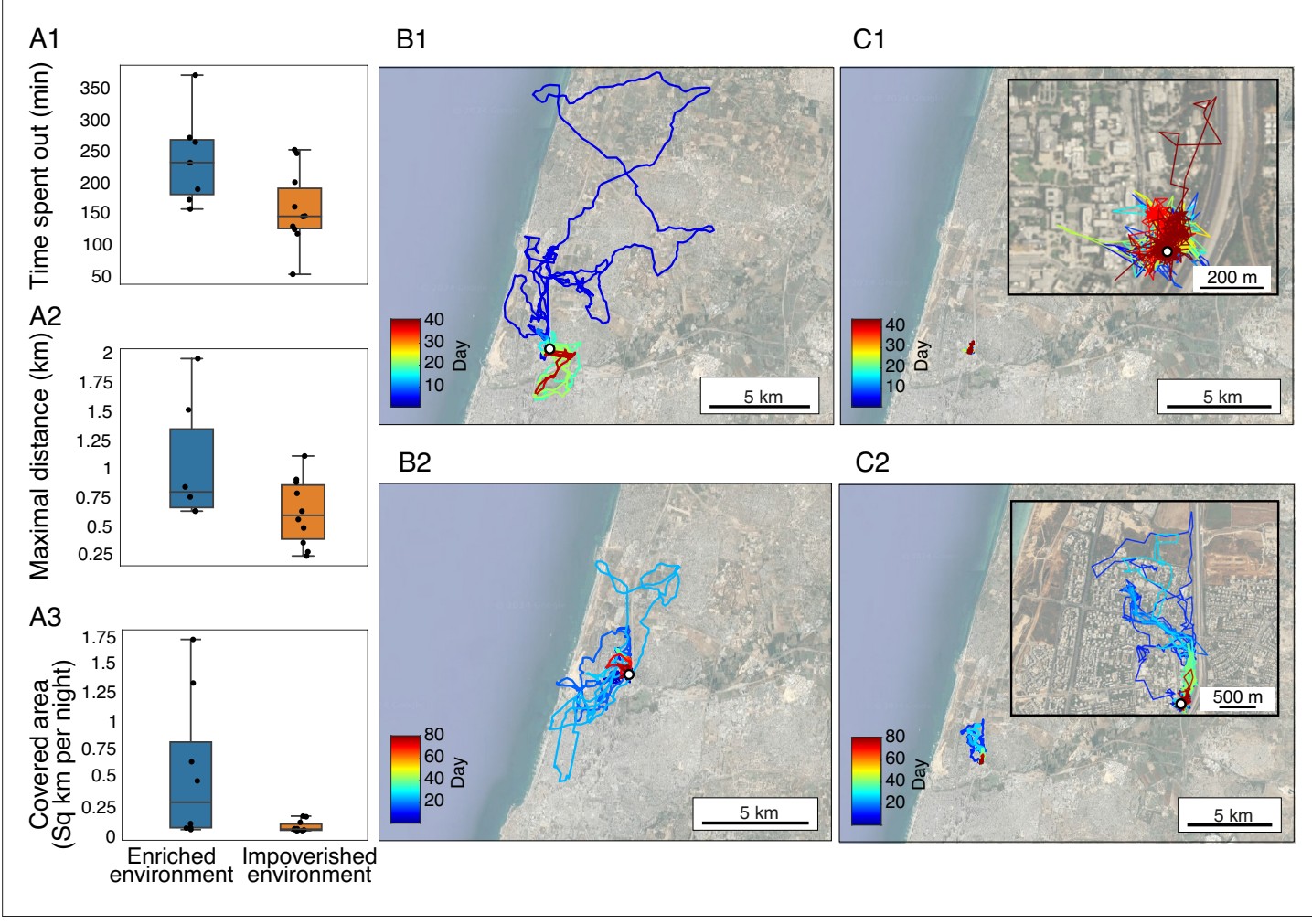

**Figure 3.** Early environmental exposure affects outdoor foraging behavior. (**A1**) Time spent by the bats outside the colony each night (in minutes). (**A2**). Distance to the furthest point from the colony per night. (**A3**) The area explored by the bats. The data for each bat were estimated for the period between its 15th and 20th days outdoors (n=17 bats). Median and whiskers based on 1.5 IQR are shown. (**B–C**) The complete movement of four individuals - two from each colony. (**B**) Individuals raised in the enriched colony, data shown for two individuals, each representing a different measured season (**B1, B2**); (**C**) Individuals raised in the impoverished colony, data shown for two individuals, each representing a different measured season (**C1, C2**). Colors depict time in days. The open colony is marked by a white circle. Insert in C1-2 zooms-in on the tracks. More examples of individual bat movement maps are presented in *Figure 3—figure supplement 1*.

The online version of this article includes the following figure supplement(s) for figure 3:

**Figure supplement 1.** Early environmental exposure affects outdoor foraging behavior.

**Figure supplement 2.** Early environmental exposure affects outdoor foraging behavior.

improved learning abilities and with the utilization of multiple cues for obtaining rewards *White and Brown, 2015*. Notably, unlike any previous study, our experimental paradigm allowed us to examine the animals' behavior in their natural habitat following manipulation of their environment early on in life.

The observed behavioral changes in the enriched bats may be the result of increased learning opportunities and enhanced problem-solving abilities driven by early-life exposure to a more dynamic environment. The enriched environment likely provided the bats with more varied experiences, allowing them to practice and refine their foraging and exploratory behaviors. These additional opportunities for learning could have contributed to the increased boldness and exploratory traits observed in the enriched bats. By encountering a more complex environment, these bats may have developed greater behavioral flexibility, enabling them to adapt more efficiently to novel situations in the wild.

**Table 4.** Mixed-effect generalized linear model (GLMM) results for outdoors measurements with the first boldness as predisposition. The lines depicting results for the environmental and the predisposition effects are highlighted in green and blue, respectively. response ~1 + Environmental_condition + Baseline_boldness + Sex + Age + Days_spent_outside + (1 | Bat_ID).

| Response | AIC | | BIC | Loglikelihood | Deviance | | | |
|---|---|---|---|---|---|---|---|---|
| | 8638.2 | | 8674.8 | –4311.1 | 8622.2 | | | |
| | Fixed effects coefficients (95% CIs) | | | | | | | |
| | Name | Estimate | SE | tStat | DF | p-value | Lower | Upper |
| | Intercept | 185.42 | 74.081 | 2.469 | 710 | 0.013 | 38.014 | 332.83 |
| | Environmental condition Impoverished | –57.736 | 26.374 | –2.189 | 710 | 0.028 | –109.52 | –5.956 |
| | Boldness of the first baseline | –32.783 | 78.474 | –0.417 | 710 | 0.676 | –186.85 | 121.29 |
| | Bat sex Male | –9.0905 | 26.014 | –0.349 | 710 | 0.726 | –60.163 | 41.982 |
| | Bat age | 0.029 | 0.217 | 0.135 | 710 | 0.892 | –0.397 | 0.456 |
| Time spent outside | Number of days spent outside until assessment day | 1.416 | 0.351 | 4.028 | 710 | 6.10E-05 | 0.726 | 2.106 |
| Response | AIC | | BIC | Loglikelihood | Deviance | | | |
| | 12437 | | 12474 | –6210.7 | 12421 | | | |
| | Fixed effects coefficients (95% CIs) | | | | | | | |
| | Name | Estimate | SE | tStat | DF | p-value | Lower | Upper |
| | Intercept | 1543.2 | 651.67 | 2.368 | 710 | 0.018 | 263.73 | 2822.6 |
| | Environmental condition Impoverished | –591.24 | 228.02 | –2.592 | 710 | 0.009 | –1038.9 | –143.57 |
| | Boldness of the first baseline | –550.64 | 649.03 | –0.848 | 710 | 0.396 | –1824.9 | 723.6 |
| | Bat sex Male | –262 | 224.84 | –1.165 | 710 | 0.244 | –703.43 | 179.42 |
| | Bat age | –0.705 | 1.888 | –0.372 | 710 | 0.708 | –4.414 | 3.002 |
| Maximum distance per night | Number of days spent outside until assessment day | 10.982 | 3.868 | 2.838 | 710 | 0.004 | 3.386 | 18576 |
| Response | AIC | | BIC | Loglikelihood | Deviance | | | |
| | 116.55 | | 123.68 | –50.277 | 100.55 | | | |
| | Fixed effects coefficients (95% CIs) | | | | | | | |
| | Name | Estimate | SE | tStat | DF | p-value | Lower | Upper |
| | Intercept | –4.31 | 6.312 | –0.682 | 12 | 0.507 | –18.065 | 9.444 |
| | Environmental condition Impoverished | –5.286 | 2.081 | –2.539 | 12 | 0.025 | –9.821 | –0.75 |
| | Boldness of the first baseline | 7.746 | 6.454 | 1.2 | 12 | 0.253 | –6.315 | 21.809 |
| | Bat sex Male | –2.354 | 2.26 | –1.041 | 12 | 0.318 | –7.279 | 2.57 |
| | Bat latest age | 0.001 | 0.022 | 0.068 | 12 | 0.946 | –0.047 | 0.05 |
| Explored area | Total amount of nights the bat spent outside | 0.183 | 0.074 | 2.466 | 12 | 0.029 | 0.021 | 0.346 |

## Indoor behavior

In contrast to the outdoor behavior, the behavioral traits that we assessed in controlled indoor experiments were not affected by the early environmental enrichment, i.e., enriched bats did not become bolder or more exploratory under laboratory conditions. When examining the bats' behavioral traits across the five trials, we found that boldness and exploration demonstrated a similar temporal dynamics, increasing between consecutive trials that were performed closely to one another in time and decreasing following a prolonged period since the previous trial (*Figure 2*). This pattern indicates that despite the bats' initial habituation to the set-up, following a period of no exposure to the

**Table 5.** Mixed-effect generalized linear model (GLMM) results for outdoors measurements with the baseline trial PC1 as predisposition.

The lines depicting results for the environmental and the predisposition effects are highlighted in green and blue, respectively.

| Response | AIC | BIC | Loglikelihood | Deviance | | | | |
|---|---|---|---|---|---|---|---|---|
| | 8636.9 | 8673.5 | –4310.4 | 8620.9 | | | | |
| | Fixed effects coefficients (95% CIs) | | | | | | | |
| | Name | Estimate | SE | tStat | DF | p- value | Lower | Upper |
| | Intercept | 161.22 | 66.908 | 2.409 | 710 | 0.016 | 29.864 | 292.58 |
| | Environmental condition Impoverished | –58.829 | 26.251 | –2.241 | 710 | 0.025 | –110.37 | –7.289 |
| | PC1 of the first baseline | 0.619 | 54.32 | 0.011 | 710 | 0.990 | –106.03 | 107.27 |
| | Bat sex Male | –7.650 | 26.159 | –0.292 | 710 | 0.770 | –59.01 | 43.708 |
| | Bat age | 0.084 | 0.242 | 0.347 | 710 | 0.727 | –0.391 | 0.559 |
| Time spent outside | Number of days spent outside until assessment day | 1.347 | 0.379 | 3.547 | 710 | 0.0004 | 0.601 | 2.093 |
| Response | AIC | BIC | Loglikelihood | Deviance | | | | |
| | 12438 | 12474 | –6210.9 | 12422 | | | | |
| | Fixed effects coefficients (95% CIs) | | | | | | | |
| | Name | Estimate | SE | tStat | DF | p-value | Lower | Upper |
| | Intercept | 1434.7 | 594.15 | 2.414 | 710 | 0.015 | 268.2 | 2601.2 |
| | Environmental condition Impoverished | –580.89 | 226.37 | –2.566 | 710 | 0.010 | –1025.3 | –136.46 |
| | PC1 of the first baseline | –365.17 | 460.75 | –0.792 | 710 | 0.428 | –1269.8 | 539.43 |
| | Bat sex Male | –318.9 | 225.1 | –1.416 | 710 | 0.157 | –760.83 | 123.03 |
| | Bat age | –1.108 | 2.121 | –0.522 | 710 | 0.601 | –5.274 | 3.057 |
| Maximum distance per night | Number of days spent outside until assessment day | 11.783 | 4.107 | 2.868 | 710 | 0.004 | 3.718 | 19.847 |
| Response | AIC | BIC | Loglikelihood | Deviance | | | | |
| | 117.82 | 124.94 | –50.909 | 101.82 | | | | |
| | Fixed effects coefficients (95% CIs) | | | | | | | |
| | Name | Estimate | SE | tStat | DF | p-value | Lower | Upper |
| | Intercept | 1.331 | 5.878 | 0.226 | 12 | 0.824 | –11.476 | 14.139 |
| | Environmental condition Impoverished | –5.544 | 2.154 | –2.573 | 12 | 0.024 | –10.239 | –0.849 |
| | PC1 of the first baseline | –1.518 | 4.323 | –0.351 | 12 | 0.731 | –10.939 | 7.902 |
| | Bat sex Male | –1.712 | 2.274 | –0.752 | 12 | 0.465 | –6.668 | 3.242 |
| | Bat latest age | –0.005 | 0.025 | –0.217 | 12 | 0.831 | –0.061 | 0.050 |
| Explored area | Total amount of nights the bat spent outside | 0.174 | 0.077 | 2.243 | 12 | 0.044 | 0.005 | 0.343 |

**Table 6.** Pearson correlation test results.

| | Pearson r value | Pearson p-value |
|---|---|---|
| Time spent out vs maximal distance | 0.28 | 0.23 |
| Time spent out vs explored area | 0.21 | 0.38 |
| Maximal distance vs explored area | 0.82 | <1.269e-05 |

set-up, the bats' boldness (or exploration) again reduced, probably due to some sort of sensitization to the set-up. The bats maintained their within-individual tendencies while doing so, suggesting that indeed boldness (and to a lesser degree exploration) were good measures of consistent personality. It is possible that a different measure of exploration (one that does not suffer from a ceiling effect like the one we used here) might have shown more consistency over time. Activity, in contrast, continuously (and significantly) showed a different pattern decreasing over time independently of the time elapsed between trials (*Figure 2*). This decrease in activity reflects familiar pups' behavior, which usually becomes less frantic over time.

The enriched bats were significantly bolder in Trial 5 (*Figure 2*). This could be a real difference indicating an interesting interaction (positive feedback) between the early exposure to an enriched environment with the more exploratory outdoors behavior. That is, the more exploratory tendency of the enriched bats made them even bolder over time. However, we could not overrule the possibility that the difference in trial 5 was a result of a post-selection bias generated by the tendency of bolder individuals in the enriched condition to remain in our colony until the end of the study, while many of the bolder ones from the impoverished condition left the colony earlier and probably moved to other colonies. This tendency can be seen in *Figure 2* when examining only the bats that had remained until the last trial (*Figure 2C1* – and *Figure 2—figure supplement 1* - see the difference in boldness in trial 2 when only examining the bats that remained in the colony). More research is needed to tell apart these two explanations.

## Outdoor tracking

The enriched bats spent significantly more time foraging outdoors, exhibited greater flight distances, and were more exploratory compared to the impoverished bats. The post condition selection bias discussed above likely did not drive these differences in outdoors behavior because there was no correlation between the indoor and outdoor behaviors. Moreover, comparisons between groups were also performed between their 15th and 20th days outdoors. This period was selected to maximize the number of individuals included before some left the colony. These differences could result in actual differences in foraging success. As fruit bats must constantly keep track of an ever-changing landscape of resources (fruit trees) (*Harten et al., 2024*), being exploratory might pose an advantage for such individuals, especially in a rapidly changing urban environment.

Interestingly, we found no significant correlation between the time bats spent outside and the maximal distance they flew per night. This finding suggests that spending more time outdoors does not necessarily reflect longer foraging journeys; some bats may have remained near the roost despite prolonged activity. This distinction highlights the importance of measuring multiple behavioral dimensions, as time spent outside alone may not accurately capture the spatial scale of foraging behavior.

Research conducted by *Egert-Berg et al., 2021*, focusing on the foraging behavior of Egyptian fruit bats in the wild, demonstrated that bats that roost in cities display more exploratory behavior than bats that roost in rural habitats. Our current study suggests how such differences in behavior could arise as a result of early-life exposure to an enriched environment, such as that in the urban environment, which is much more diverse and dynamic than the rural environment. Urban roosting bats encounter more types of food and must practice more diverse skills to obtain it, like the bats in our enriched colony. Our findings can thus explain how bats and other animals with early exposure to the urban environment become more exploratory. Although fruit-bat pups tend to be more active than adults, it is possible that such adaptations can also be acquired by the adults as well.

The lack of correlation found between the indoor and outdoor behaviors emphasizes the significance of assessing behavior in the bats' natural habitat to obtain more reliable and meaningful results. While measuring personality traits, particularly boldness and exploration, in a small captive setup clearly assesses some aspects of personality, translating these to behavior in the real world may be limited. It is possible that other behavioral traits measured indoors, such as neophobia or decision-making under uncertainty, might have shown stronger associations with outdoor behavior. Although some studies have found an indoor/outdoor correlation in other animals (see *Herborn et al., 2010*), our findings highlight the importance of choosing behavioral traits that are ecologically relevant, and suggest that the traits we measured may not have captured the dimensions most predictive of natural behavior.

**Table 7.** Egyptian Fruit bat pups were captured with their mothers in rural and urban colonies and then divided between two environmental conditions, enriched and impoverished.

| Season | Environmental condition | Colony ID | Coordinates | Number of pairs |
|---|---|---|---|---|
| | | Tinshemet | 31°59'43.2"N 34°57'19.2"E | 1 |
| | | Beit Guvrin | 31°36'45.8"N 34°53'41.7"E | 1 |
| | Enriched | Herzliya | 32°10'18.4"N 34°48'51.1"E | 4 |
| | | Tinshemet | 31°59'43.2"N 34°57'19.2"E | 1 |
| | | Beit Guvrin | 31°36'45.8"N 34°53'41.7"E | 1 |
| 1 (2019–2020) | Impoverished | Herzliya | 32°10'18.4"N 34°48'51.1"E | 3 |
| | | Beit Guvrin | 31°36'45.8"N 34°53'41.7"E | 5 |
| | Enriched | Herzliya | 32°10'18.4"N 34°48'51.1"E | 9 |
| | | Beit Guvrin | 31°36'45.8"N 34°53'41.7"E | 5 |
| 2 (2020–2021) | Impoverished | Herzliya | 32°10'18.4"N 34°48'51.1"E | 10 |

Overall, our study provides evidence of the profound impact of early life environmental conditions on the behavior of the adult bats. These findings contribute to our understanding of the developmental factors that shape animal behavior and highlight the importance of environmental enrichment during the early life stage.

## Methods

### Experimental animals

The experiment was conducted on young Egyptian fruit bats. A total of 40 bats were captured from three wild colonies (Herzliya, Beit Guvrin, and Tinshemet; *Table 7*) between the years 2019–2021 and housed in the Zoological Garden at Tel Aviv University. We captured pregnant females or females with young pups before their volant stage to ensure that the pups would be naïve and without previous individual flight experience. Of the 40 bats in the study, 21 were born in our lab and 19 were captured at a very young age together with their mothers, before they could fly independently. To avoid an origin bias, bats from all colonies were divided equally and housed in two identical rooms (~2.5×2×2.5 m³, *Table 7*), maintained at a controlled temperature of 24–26°C and a light cycle approximately matching the times of sunrise/sunset. The bats' diet in both environments comprised seasonal mixed fruits (watermelon, apples, melons, and bananas *ad lib*).

Upon arrival, the bats' weight and forearm were measured and documented. Each mother-pup pair was bleached with a unique mark, and each bat was also tagged with a subcutaneous electronic RFID chip. All bats were measured and checked regularly every 2-3 weeks to make sure they were healthy. Three individuals were removed from the experiment: two due to injury (broken wings), and one due to weight loss (>10% loss of their previous body weight).

### Environmental conditions

While both colonies were provided with the same basic conditions and food, the bats in the 'enriched' environment regularly experienced varying environmental enrichment three times a week, including changing the number and shape of food bowls, hanging food baskets from the ceiling, playback of vocalizations recorded from a larger colony, and adding rope ladders or ropes hanging from the ceiling. In the 'impoverished' environment, the animals experienced a dull environment with no variation in conditions throughout the study period. They received their food in one bowl, while nothing was hung from the ceiling and no other enrichment was provided.

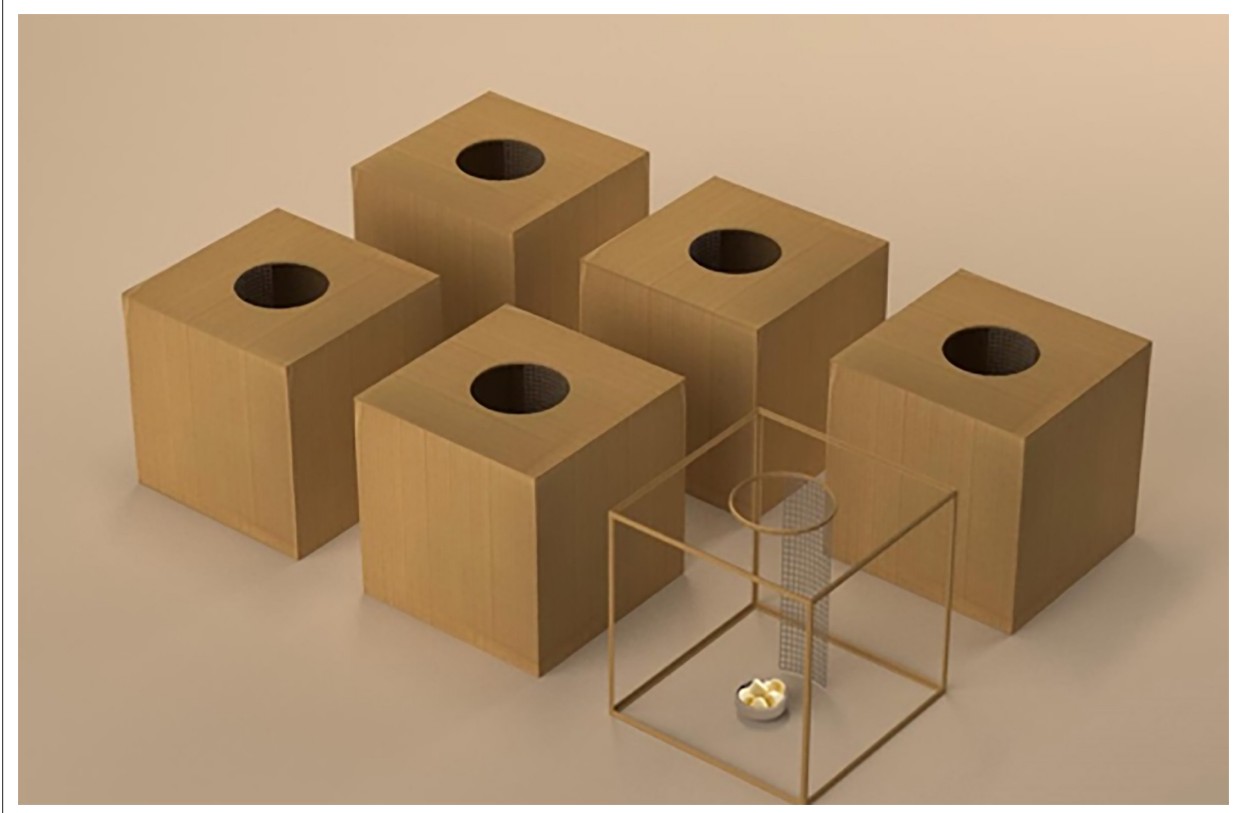

**Figure 4.** Experimental set-up schematic. Each bat is placed in a tent with six foraging boxes.

### Defining baseline personality in juveniles – the foraging box experiment

As juveniles (55-281 days old), the young bats participated in a foraging box experiment to examine a set of personality traits in a captive setting (*Harten et al., 2021*). The experimental set-up consisted of an indoor tent (3.9×2.7×1.9 m³), with six evenly spaced black plastic boxes placed on the floor (i.e. foraging box, 64×38×40 cm³). Each box had a round hole in its lid (10 cm diameter) with a mesh ladder leading down to a food bowl, containing a sufficient amount of food for the bat for an entire night, to prevent the bat from selecting another box simply because the first one is empty (*Figure 4*).

#### The exposure phase

Bats are social animals (*Kerth, 2008*; *Kwiecinski and Griffiths, 1999*). To reduce the stress of being alone in a new environment before the first trial, a group of 4-5 pups from the same environment was placed in the tent overnight (16:00-8:00) for a session of exposure. Each of the six boxes contained an amount of food sufficient for all of the bats taking part in the exposure.

**Table 8.** Number of bats that participated in each of the trials.

|  | Environmental condition | Baseline (both 1+2) | Post-enrichment trial | Post-release trials (4+5) |
|---|---|---|---|---|
| Season 1 | Enriched | 6 | - | 4 |
|  | Impoverished | 5 | - | 5 |
| Season 2 | Enriched | 13 | 14 | 5 |
|  | Impoverished | 15 | 15 | 5 |
| Total number of bats |  | 39 | 29 | 19 |

**Table 9.** Full timeline of the experiment.

| | | October-December | | | December-June | | |
|---|---|---|---|---|---|---|---|
| 2019–2020 | Baseline Trials n=11 | Environmental enrichment | | Release to open colony | Tracking bats using GPS devices n=10 | Post-release trials n=9 |
| 2020–2021 | Baseline Trials n=29 | Environmental enrichment | Post-enrichment trial n=29 | Release to open colony | Tracking bats using GPS devices n=11 | Post release trials n=10 |

### The experiment

Following this one night of introduction, each pup (n=39, see *Table 8*) was placed alone in an open carrying bag (35×26×30 cm³) inside the flight tent with the six foraging boxes (described above) for a full night. Each food bowl offered an amount of food that was enough for the full night, containing 25 pieces of fruit (with a total weight of 150 g) and 50 ml of mango nectar. The experiment took place two nights in a row to determine behavioral consistency. The experiment was recorded using an IR video camera (Sony HDRCX730, Sony FDR-AX53), together with one infrared light (Methaphase Technologies Inc-ISO-14- IR-24) placed inside the tent.

The environmental enrichment began only after all the bats in the enriched room had completed the baseline experiments (see *Table 9* for the full timeline of the experiment). Both groups then experienced the (enriched/impoverished) environment for at least two months before moving on to the next stage, the open colony, from which the bats are free to fly in and out as they choose (see details below).

During the second season (year 2020-21), following the exposure to the environmental conditions, all bats (n=29, *Table 8*) underwent an additional night in the foraging box experiment to evaluate the effect of the enriched environment. This additional trial took place prior to their transfer to the open colony and is termed the 'post-enrichment trial'.

### Personality after foraging in the wild

Bats that routinely exited the open colony to forage and returned were twice tested again in the foraging box experiment: 2-3 weeks following their first exit (Trial 4, post-release 1), and 5-6 weeks following their first exit (Trial 5, post-release 2). A total of 19 bats were tested for these trials (see *Table 8*).

### Behavioral analysis

Each bat's actions during the foraging box experiment were documented based on video recordings by two independent observers. In cases of disagreement, a third observer compared the two and made the final judgment. The identified actions comprised: landing on or entering the carrying bag in which the bats had been introduced into the room; landing; entering the food boxes; collecting food; or flying around to explore the environment.

Similar to our previous study (*Harten et al., 2021*), we then calculated the following personality traits:

**Boldness/Risk-taking:** Boldness can be defined in several ways (*Carter et al., 2013*). We measured it as the proportion of times the bat entered the boxes out of the total number of landings and entering with or without collecting food from the boxes. A previous study has shown that entering boxes placed on the ground is a risky behavior that a bat generally avoids (*Harten et al., 2021*).

**Exploration:** The exploratory index was calculated as the number of individual foraging boxes the bat visited (i.e. 0-6). Note that each box contained enough food for the entire night eliminating the need for it to visit more than one box.

**Activity:** Activity level was defined as the number of interactions with the boxes (landings and entries) normalized by the total time in the experiment (i.e. after the bat had left the carrying bag for the first time).

### Statistical analysis of the behavioral trials

To verify that there were no behavioral differences between the two groups prior to applying our environmental conditions, we compared the baseline trials between the two environmental conditions (*Supplementary file 1*). We used a mixed-effect general linear model (GLMM) implemented

in MATLAB R2019, with the traits set as the response variable, the environmental condition and trial number (baseline 1 or 2) set as fixed factors, and bat ID set as a random effect (n=39, *Table 8*).

We use the term 'personality' to encompass a set of behavioral traits that exhibit consistency within individuals over time and across various contexts (*Stamps and Groothuis, 2010*). To evaluate the consistency of the measured behavioral traits, we conducted a comparison between the bats' first trial (baseline 1) and the trial conducted following their exposure to the different environmental conditions (post-enrichment trial). These analyses incorporated only the individuals from the second measuring season, because we only ran the post-enrichment measurement during the second season.

To assess the impact of the environmental condition on each behavioral trait, we employed GLMMs. We used the behavioral trait as the response parameter, the tested environmental condition (i.e. enriched or impoverished), trial number (1-5), and the interaction between environmental conditions and trial number as fixed factors. Bat ID was set as a random effect.

## Analyzing the bats' outdoor foraging behavior

Following the captive stage described above, bats were transferred to our open colony, an Egyptian fruit bat colony, representing a natural urban roost where bats can exit nightly to forage in the wild and return to roost in our fully monitored day-roost (or move to a different roost). Food is provided in the roost every evening. This system allows us to continuously monitor the bats' natural behaviors within and outside the roost (*Harten et al., 2020*).

The bats were released in groups of 4-6 individuals (at an average age of 218.92±56.8 days) from the same experimental colony (either the enriched or impoverished environment), alternately between these two environmental conditions. Each bat was fitted with an on-board GPS device (Vesper, ASD Inc) following its second foraging nights in the wild, in order to monitor most of the navigational history of all individuals. The device was coated using Parafilm (Heathrow Scientific) and duct tape, and was attached to the bats using a chain necklace covered with heat-shrink (see full details in *Goldshtein et al., 2022*). The mean weight mounted on the bats was 6 g, accounting for 5.6±0.6% of the bats' weight. The GPS was programmed to start recording a few hours before sunset and stopped at 6:00 AM, collecting GPS locations every 30 s.

These data allowed us to assess various parameters extracted from the individuals' natural foraging behavior, which served as proxies for their exploratory and activity tendencies. Specifically, we measured the:

**Maximum flying distance:** Defined as the distance to the furthest point from the open colony that the bat reached on every night outside.

**Time spent outdoors:** Defined as the total time the bat spent outside the colony on a given night. Nights when a bat did not leave the colony (6.2 days per bat on average) were not included in the analysis.

**The explored area** was estimated as follows: For each bat, we calculated the maximum distance from the colony for each night. We then computed the 95th percentile of these nightly maximum distances and used it as a threshold radius. Using the colony as the center point, we excluded all GPS locations falling beyond this radius. A convex hull was then calculated from the remaining points to estimate the core area utilized by each bat.

Statistics: To examine the influence of the environmental condition (i.e. enriched or impoverished), we used GLMs for each of the three outdoor traits, with the traits set as the response parameter. Environmental condition, bat age and sex, and the number of nights the bat spent outside until the assessment day were set as a fixed effect. For the explored area analysis, the total number of days the bat spent outside during the entire tracking period and the bat's age at the last time of measuring were set as fixed effects. For the maximum distance and the time spent outdoors, we used data from each night the bat was outside. Explored area receiving only one (seasonal) measurement for each bat.

## Captivity and outdoor comparison

We sought to demonstrate the effectiveness of our captive setup by comparing the individual's behavior in both indoor and outdoor settings. To achieve this, we analyzed the Pearson correlation between the first post-release trial and the average of the subsequent 5 days of outdoor measurements using Pearson correlation test. In addition, we tested the correlation between the individual's first post-release trial and the average of its subsequent 20 days of outdoor measurements.

## Acknowledgements

We would like to thank Mor Taub for assistance with figure design, Goni Naamani for assisting with the data analysis. This project was partially funded by the ERC project BehaviorIsland.

## Additional information

### Funding
No external funding was received for this work.

### Author contributions
Adi Rachum, Data curation, Formal analysis, Writing - original draft; Lee M Harten, Methodology; Reut Assa, Nesim Gonceer, Data curation; Aya Goldshtein, Writing – review and editing; Xing Chen, Formal analysis; Yossi Yovel, Supervision

### Author ORCIDs
Adi Rachum ⓘ https://orcid.org/0009-0007-2494-0602
Yossi Yovel ⓘ https://orcid.org/0000-0001-5429-9245

### Ethics
Experiments were approved by the TAU IACUC; permit number: 04-18-030. Capture of Egyptian fruit bats for the research was approved by the Israel National Park Authority, permit number 2020/42646.

Reviewer #1 (Public review): https://doi.org/10.7554/eLife.103220.3.sa1
Author response https://doi.org/10.7554/eLife.103220.3.sa2

## Additional files

### Supplementary files
Supplementary file 1. Mixed generalized linear model (GLM) results for baseline trials of two environmental conditions response ~1 + Environmental_condition + Trial_number + (1| Bat_ID).

Supplementary file 2. Mixed-effect generalized linear model (GLMM) results for proportion of nights foraging outside the roost (%). DaysOutProportion ~1 + EnvironmentalCondition + Sex + Age + (1|Bat_ID) With Y ~ Binomial (TotalExperimentDays) and link = logit.

MDAR checklist

### Data availability
The datasets generated and analyzed during the current study are available on Mendeley Data: https://doi.org/10.17632/wh7c636y3t.1.

The following dataset was generated:

| Author(s) | Year | Dataset title | Dataset URL | Database and Identifier |
|---|---|---|---|---|
| Rachum A, Harten L, Assa R, Goldshtein A, Chen X, Gonceer N, Yovel Y | 2024 | Data for : Early experience affects foraging behavior of wild fruit-bats more than their original behavioral predispositions | https://doi.org/10.17632/wh7c636y3t.1 | Mendeley Data, 10.17632/wh7c636y3t.1 |

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
