## [Editor Report · eLife Assessment]

This paper provides **important** insight into how early life experience shapes adult behavior in fruit bats. The authors raised juvenile bats either in an impoverished or enriched environment and studied their foraging behaviors. The evidence is **convincing** that bats raised in enriched environments are more active, bold, and exploratory. The work will be of interest to ethologists and developmental psychologists.

---

## [Referee Report · Reviewer #1 (Public review)]

Summary:

The authors show that early life experience of juvenile bats shape their outdoor foraging behaviors. They achieve this by raising juvenile bats either in an impoverished or enriched environment. They subsequently test the behavior of bats indoors and outdoors. The authors show that behavioral measures outdoors were more reliable in delineating the effect of early life experiences as the bats raised in enriched environments were more bold, active and exhibit higher exploratory tendencies.

Strengths:

The major strength of the study is providing a quantitative study of animal "personality" and how it is likely shaped by innate and environmental conditions. The other major strength is the ability to do reliable long term recording of bats in the outdoors giving researchers the opportunity to study bats in their natural habitat. To this point, the study also shows that the behavioral variables measured indoors do not correlate to that measured outdoor, thus providing a key insight into the importance of test animal behaviors in their natural habitat.

Weaknesses were in the first round of review:

It is not clear from the analysis presented in the paper how persistent those environmentally induced changes, do they remain with the bats till end of their lives.

Comments on revisions:

The authors have addressed those weaknesses and the paper is much stronger.

---

## [Author Response]

The following is the authors’ response to the original reviews.

**Reviewer #1 (Public Reviewer):**
It is not clear from the analysis presented in the paper how persistent those environmentally induced changes, do they remain with the bats till the end of their lives.

Currently, the long-term effects of enrichment on the bats remain uncertain. Preliminary results suggest that these differences may persist throughout the bats’ lifetimes; however, further data analysis is ongoing to determine the extent of these effects. We also addressed now at the manuscript discussion

**Reviewer #2 (Public Reviewer):**
(1) Assessing personality metrics and the indoor paradigm: While I applaud this effort and think the metrics used are justified, I see a few issues in the results as they are currently presented:(a) [Major] I am somewhat concerned that here, the foraging box paradigm is being used for two somewhat conflicting purposes: (1) assessing innate personality and (2) measuring changes in personality as a result of experience. If the indoor foraging task is indeed meant to measure and reflect both at the same time, then perhaps this can be made more explicit throughout the manuscript. In this circumstance, I think the authors could place more emphasis on the fact that the task, at later trials/measurements, begins to take on the character of a "composite" measure of personality and experience.

Personality traits should generally be stable over time, but personality can also somewhat change with experience. We used the foraging box to assess individual personality, but we also examined the assumption that what we are measuring is a proxy of personality and hence is stable over time. We now clarify this in the manuscript.

(b) [Major] Although you only refer to results obtained in trials 1 and 2 when trying to estimate "innate personality" effects, I am a little worried that the paradigm used to measure personality, i.e. the stable components of behavior, is itself affected by other factors such as age (in the case of activity, Fig. 1C3, S1C1-2), the environment (see data re trial 3), and experience outdoors (see data re trials 4/5).

We found that boldness was the most consistent trait, showing persistence between trials 1 to 5, i.e., 144 days apart on average. We thus also used Boldness as the primary parameter for assessing the effects of personality on the outdoors behavior. While we evaluated other traits for completeness, boldness was the only one that consistently met the criteria for personality, which is why we focused on it in our analyses. The other traits which were not stable over time could be used to assess the effects of experience on behavior

Ideally, a study that aims to disentangle the role of predisposition from early-life experience would have a metric for predisposition that is relatively unchanging for individuals, which can stand as a baseline against a separate metric that reflects behavioral differences accumulated as a result of experience.I would find it more convincing that the foraging box paradigm can be used to measure personality if it could be shown that young bats' behavior was consistent across retests in the box paradigm prior to any environmental exposure across many baseline trials (i.e. more than 2), and that these "initial settings" were constant for individuals. I think it would be important to show that personality is consistent across baseline trials 1 and 2. This could be done, for example, by reproducing the plots in Fig. 1C1-3 while plotting trial 1 against trial 2. (I would note here that if a significant, positive correlation were to be found (as I would expect) between the measures across trial 1 and 2, it is likely that we would see the "habituation effect" the authors refer to expressed as a steep positive slope on the correlation line (indicating that bold individuals on trial 1 are much bolder on trial 2).)

We agree and thus used boldness which was found to be stable over five trials (three of which were without external experience). We note that if Boldness as we measured it increased over time, the differences between individuals remained similar and this is what is expected from personality traits measured in the same paradigm several times (after the animal acquires experience).

(c) Related to the previous point, it was not clear to me why the data from trial 2 (the second baseline trial) was not presented in the main body of the paper, and only data from trial 1 was used as a baseline.

We added a main figure, showing the correlation between the two baseline trials

In the supplementary figure and table, you show that the bats tended to exhibit more boldness and exploratory behavior, but fewer actions, in trial 2 as compared with trial 1. You explain that this may be due to habituation to the experimental setup, however, the precise motivation for excluding data from trial 2 from the primary analyses is not stated. I would strongly encourage the authors to include a comparison of the data between the baseline trials in their primary analysis (see above), combine the information from these trials to form a composite baseline against which further analyses are performed, or further justify the exclusion of data as a baseline.

We had no intention of excluding data from baseline 2. As we have shown several times before (e.g., Harten, 2021) bats’ boldness as we measure it in the box experiment increases over sessions performed nearby in time. This means that trial 2’s boldness was higher than that of trial 1 and trial 3 which made the data less suitable for a Linear model. Moreover, our measurement of boldness is capped (with a maximum of 1) again making it less suitable for a Linear model. However, following the reviewer’s question we now ran all analyses with trial 2’s data included and not only that the results remained the same, some of the models fit better (based on the AIC criterion). We added this information to the revised manuscript.

(2) Comparison of indoor behavioral measures and outdoor behavioral measures Regarding the final point in the results, correlation between indoor personality on Trial 4 and outdoor foraging behavior: It is not entirely clear to me what is being tested (neither the details of the tests nor the data or a figure are plotted). Given some of the strong trends in the data - namely, (1) how strongly early environment seems to affect outdoor behavior, (2) how strongly outdoor experience affects boldness, measured on indoor behavior (Fig. 1D) - I am not convinced that there is no relationship, as is stated here, between indoor and outdoor behavior. If this conclusion is made purely on the basis of a p-value, I would suggest revisiting this analysis.

We agree that the relationship between indoor personality measures and outdoor foraging behavior is of great interest and had expected to find some correspondence between the two. To test this, we conducted multiple GLM analyses using the different indoor behavioral traits as predictors of outdoor behaviors. These analyses did not reveal any significant correlations. We also performed a separate analysis using PC1 (derived from the indoor behavioral variables) as a predictor, and again found no significant associations with outdoor behavior.

We were indeed surprised by this outcome. It is possible that the behavioral traits we assessed indoors (boldness, exploration, and activity) do not fully capture the dimensions of behavior that are most relevant to foraging in the wild. For example, traits such as neophobia or decisionmaking under risk, which we did not assess directly, may have had stronger predictive value for outdoor behavior. We now highlight this point more clearly in the Discussion and acknowledge the possibility that alternative or additional personality traits might have revealed meaningful relationships.

(3) Use of statistics/points regarding the generalized linear models While I think the implementation of the GLMM models is correct, I am not certain that the interpretation of the GLMM results is entirely correct for cases where multivariate regression has been performed (Tables 4s and S1, and possibly Table 3). (You do not present the exact equation they used for each model (this would be a helpful addition to the methods), therefore it is somewhat difficult to evaluate if the following critique properly applies, however...)The "estimate" for a fixed effect in a regression table gives the difference in the outcome variable for a 1 unit increase in the predictor variable (in the case of numeric predictors) or for each successive "level" or treatment (in the case of categorical variables), compared to the baseline, the intercept, which reflects the value of the outcome variable given by the combination of the first value/level of all predictors. Therefore, for example, in Table 4 - Time spend outside: the estimate for Bat sex: male indicates (I believe) the difference in time spent outside for an enriched male vs. an enriched female, not, as the authors seem to aim to explain, the effect of sex overall. Note that the interpretation of the first entry, Environmental condition: impoverished, is correct. I refer the authors to the section "Multiple treatments and interactions" on p. 11 of this guide to evaluating contrasts in G/LMMS: https://bbolker.github.io/mixedmodelsmisc/notes/contrasts.pdf

We are not certain we fully understand the comment; however, if our understanding is correct, we respectfully disagree. A GLM analysis without interaction terms—as conducted in our study—functions as a multiple linear regression, wherein each factor's estimate reflects its individual effect on the dependent variable. For example in the case of sex, it examines he effect of sex on the tie spent out independently of enrichment. An interaction term would be needed to test sex*enrichment. We have added the models’ formula, and we hope this clarifies our approach

**Reviewer #1 (Recommendations for the authors):**
I would recommend the following:(1) As video tracking and behavioral analysis softwares are wide spread, it would be great to see this applied to the bat behavior indoor to answer questions like how does the bat velocity or heading or acceleration correlate with the behavioral measures boldness , activity or exploration? In the same gist, can one infer boldness, activity or exploration from measured bat velocity or other parameters? I think this will further make the indoor behavior more quantitative.

In a tent of the size used in our study, bats’ flight behavior tends to be highly stereotypical: they typically perch on the wall, take off, circle the tent—sometimes multiple times—and then either land or not, and enter or not. Flight velocity is largely determined by individual maneuverability and the physical constraints of the space; thus, precise tracking is unlikely to provide further insight into boldness. In contrast, decision-making behaviors—such as whether to land or enter—more accurately reflect personality traits, as we have shown previously (Harten et al., 2018). Moreover, accurate 3D tracking in such an environment is possible but definitely not easy due to the many blind-spots resulting from the cameras being inside the 3D volume. Nonetheless, we quantified flight activity and assessed its correlation with the other behavioral axes. As it was highly correlated with general activity, we did not include it as an independent parameter in the main analysis. However, in response to the reviewer’s suggestion, we now present this analysis in the Supplementary Materials.

(2) It is not clear whether the bats come from the same genetic background. they might be but it is not mentioned in the methods under the experimental subjects.

We have shown in the past that there is no familial relations in a randomly caught sample of bats in the colony where we usually work (Harten et al., 2018). The bats were caught in three, not related wild colonies. The text referring to the table was clarified in the revised manuscript

(3) It will be great to include the author's thoughts about mechanisms underlying those environmentally induced changes in behavior in the discussion section along with how this will affect the bats' social foraging abilities. Another question that comes to mind is whether growing up with a large number of bats constitute an enriched environment in itself.

We agree that this could count as an enrichment, and we thus ensured similar group sizes in both groups for this reason. We clarify this in the revised manuscript.

We have elaborated on the underlying mechanisms in the discussion, focusing on how they contribute to behavioral changes.

**Reviewer #2 (Recommendations for the authors):**
(1) Outdoor foraging behaviorIf I understand correctly, the data you display in Fig. 3A is only from the 2nd to 3rd weeks of exploration, i.e. just before the first post-exploration trial.What does the data look like for the second outdoor exploration data, i.e. before the final trial?Is there a specific reason why these measures were only computed on the GPS data from the 3rd week outside? If so, can this sampling of the data be motivated or briefly addressed (in the methods and wherever else necessary)?

In order to allow a comparison between individuals, we had to restrict ourself to a period we had data from many individuals (some dissapeared later on).

Following the reviewer suggestion – we added a supplemenry figure including days 21-26

I would find it important and of great interest to see movement maps for more animals, as these give very rich information that is not entirely captured by the three proxies of outdoor activity.Are these four exemplary animals sampled from both seasons?Did you check to see if there were any overall differences in outdoor foraging behavior as a function of the season in which the bats were captured?

Yes, the samples represent individuals from both tested years. This was clarified, and additional examples were included in a supplementary figure.

Variable of time spent outdoors: You mention that you did not include the nights that the bat spent in the colony in these calculations. Did you also look to see if 'the number of nights when the bats left the colony' predicted the bat's earlier enrichment treatment? This could also be interesting to consider.

In response to the reviewer’s comment, we conducted an additional analysis to test whether the proportion of nights each bat spent foraging outside the roost was predicted by its earlier environmental condition (enriched vs. impoverished). We also examined whether sex or age influenced this variable. This analysis showed no significant effect of environmental condition, sex, or age on the proportion of nights spent foraging outside the roost

[Following on point 3 in public review...]When wishing to discuss the effect/significance of predictors overall, it is common to present the modelling results as an analysis of variance table. See, for example, the two-way anova section (p. 182) in the book Practical Regression and ANOVA using R: https://cran.r-project.org/doc/contrib/Faraway-PRA.pdfI think the output of passing the model object to an "anova" yields the table that you may be looking for, where the variance accounted for by a predictor is given overall, and not just relative to the first level of all predictors. Naturally, this information can be used in combination with the information provided by the raw model output presented in the paper.I assume you have done this analysis in R, but am not sure, as the statistical software used is not mentioned. There are several packages in R that allow users to quickly plot the graphical interaction of the parameters they use in models, which aids in interpreting results. It would be good to check results of model fitting in this manner.Relatedly, I was unable to locate the data and code for this paper using the DOI provided. Neither searching the internet using the doi nor entering the doi on the Mendeley Data website returned the right results. I tried searching Mendeley Data using the senior author's last name, but the most recent entry does not appear to be from this paper. https://data.mendeley.com/datasets/fr48bmnhxj/1

We thank the reviewer for the helpful comment. The analysis was indeed conducted in MATLAB, and this has now been clarified in the manuscript. We have also revised the result tables to improve clarity and included the exact formulas used for each model. Regarding the data availability, the reviewer is correct — the dataset had not yet been published at the time of submission. It is now available at the provided DOI link.

### Suggestions and questions for the present paper, grouped thematically:[Major] Expansion and development of results: I thought there were many interesting and suggestive points in this data that could be expanded upon. I mention some of these here. While the authors of course do not need to implement all of these suggestions, I think the paper would benefit from a more substantial presentation of this rich data set:(a) Individual differences as such are not emphasized in the paper so much, as the analyses, particularly those expressed as boxplots, are grouped. The scatter plots in Figure 1 give the richest insight into how individual behavior changes throughout the course of the experiment. I would advocate for the authors to show additional comparisons using such scatter plots (perhaps in the supplementary, if needed).

We thank the reviewer and added scatter plots to figure 2

(b) In the second paragraph of the results, the authors introduce the concept of a pareto front and that of personality archetypes (lines 101-107). I found this very interesting, but these concepts were never reiterated upon later in the results or in the discussion. In fact, at many points, I found myself curious as to how the three indoor measures of personality might be combined to form a composite measure of personality (and likewise for outdoor measures). Have you tried to combine measures into a composite and tried to measure whether this composite metric provides any additional insight into these phenomena? For example, what if you mapped the starting position of each bat as a point in a three-dimensional space, given by the three personality measures, and then evaluated their trajectory through this space with measurements taken at later trials. Could innate personality be interpreted as the starting vector in this space (measured across the two baseline trials)?

Following the reviewer’s (justified) curiosity we ran a PCA analysis on the behavioral data from trials 1 and 5 and found that there is a significant correlation between the individual scores on PC1. This can be thought of as a measurement that takes both boldness and exploration into account (the weight of activity was very low). We added this information to the revised manuscript and also use this new behavioral parameter as a predisposition in the models (instead of exploration and activity).

Could environmental exposure be quantified as a warping of the trajectory through this space? Finally, could outdoor experience also be incorporated to evaluate how an individual arrives at its final measurement of personality combined with experience (trial 5)?The paper currently tries to explain outdoors behavior given personality and not vice versa. While this is a very interesting suggestion, we feel that adding this analysis would make the premise of the paper less clear and since the paper is already somewhat complex, we prefer to leave this analysis for a future study.

Examining the 3D trajectories of the individuals through the personality space did not reveal any immediate clear pattern (triangles mark the first trial and colours depict the environmental treatment) –

**Author response image 1. sa2fig1:** 

Related to this point: I think the strongest part of the paper is the result showing that bats exposed to enriched environments explore farther, more often, and over larger distances than bats that were raised in an impoverished environment.

We completely agree and tried to further emphasize this

(c) While these results of the outdoor GPS tracking are very clear, I wish that more information were extracted from the tracking data, which is incredibly rich and certainly can be used to derive many interest parameters beyond those that the authors have shown here. Examples might include: distance travelled (as opposed to estimated km2 or farthest point), a metric of navigational ability (how much "dead reckoning" the animal engages in). I even wonder if the areas or landmarks visited by the enriched bats might be found to be more complex, challenging, or richer by some measure.

This study was a first step, aiming to establish a connection between early exposure and outdoors foraging

We agree that there are many more analyses that can be done and indeed that ones related to navigation capabilities are missing. We are still collecting data on these bats and hope to present a more advanced analysis with a time span of years.

(d) Related to the above point: I find it very interesting that in 3 of the 4 bats for which you show exemplary movement data (Fig. 3, panels B and C), they appear to travel to the farthest distances and cover the most ground early on, and become more "conservative" in their flight paths on later evenings. This point is not explored in the discussion, nor related to earlier measurements.

During the first months of exploration, bats will occasionally perform long exploratory flights in between bouts of shorter flights where they return to nearby familiar trees. This behavior can be seen in more detail in Harten et al Science 2020. We are currently quantifying this more carefully for another study.

(e) Finally, my points about the possible strength of a composite measure of the three personality metrics is related to my concern about one of the conclusions, which is that innate personality does not have an effect on outdoor foraging behavior. I think the manner in which this was tested statistically is likely to bias the results against finding such a result given that personality metrics are used to predict outdoor behaviors in an individual manner (6 models in total, each examining a single comparison of predisposition to outdoor behavior), while both indoor personality metrics (Fig 1B) and outdoor behaviors appear to be correlated with each other (Table 5).Are there other analyses you have performed that are not presented in the paper and that have led you to conclude that there is no relationship here?

We agree with the reviewer, that our findings do not exclude an effect of innate personality on foraging but only suggest no such affect for the parameter we measured. That said, we did expect to find an effect of boldness because this parameter has been shown to differentiate much between groups (Harten et al., 2018), and to correlate with other parameters of behavior. We were therefore surprised to find no significant effects, as we had anticipated observing some differences.

Following the reviewer’s previous comment we now also tested another predisposition parameter – the PC1 score and also found that it did not explain foraging.

(f) Personality measured before and after early environmental exposure (related to point (a) above): I find it interesting that the positive correlation in boldness between baseline and post-enrichment or baseline and post-release suggests that the individuals that were the most bold remained bold (and likewise for less adventurous individuals). The correlation for activity, too, still suggests that more active individuals early in life are likely to remain very active after enrichment, even accounting for the fact that activity is confounded with age.Perhaps you could place some emphasis on the fact that the initial variation between individuals also appears to be relatively stable over repeated trials. You might also consider measuring this directly (population variance over successive trials; relationship of population variance on indoor measures vs. outdoor measures...)

Yes – this is a main point of interest. We further emphasize that in the revised manuscript

(g) Effect of indoor behavior following early experience on outdoor behavior: You evaluate the effect of predisposition (measured on baseline trial 1) and environmental condition on measures of outdoor activity (Table 4). I wonder if you also tried using indoor behavioral measures measured on the post-enrichment trial 3 to predict outdoor foraging behavior.Assuming that these measures are in fact reflecting a combination of predisposition and accumulated experience, then measurements at this closer time point may tell you how the combination of innate traits and early acquired experience affect behavior in the wild.

We appreciate the reviewer’s insightful suggestion to test whether indoor behavior from post-enrichment Trial 3, reflecting both innate traits and experience, predicts outdoor foraging behavior. We conducted this analysis, but found that the boldness in Trial 3 did not significantly predict any of the outdoor activity measures.

(2) [Minor] Age/development: While the authors discuss the effect of their manipulations on behavioral measures, they do not much discuss the effect of age.I think it would be important to include at some point a mention of the developmental stages of Rousettus, giving labels to certain age ranges, e.g. pup, juvenile, adult, and to provide more context about the stages at which bats were tested in the discussion. Presently, age is only really mentioned as an explanation for declining activity levels, but I wonder if it might also have an influence on boldness.It would also be very elegant for figures where age is given in days, to additional label then with these stages.

All bats were juveniles during the trials (approximately 4 to 8 months old), so they could not be divided into distinct age groups. To assess the effect of age, it was included as a predictor (in days) in the GLM analysis.

(3) [Major] Effect of early experience and outdoor experience on the indoor task: In the paragraph on lines 278-285, you argue that the effect of seeing earlyenriched bats exhibit more boldness in trial 5 was likely due to post-sampling bias...I tend to disagree with this conclusion. I actually find this result both interesting and intuitive - that bats that were exposed to an enriched environment and have had experience in the wild, show much bolder activity on a familiar indoor foraging test (i.e. outside experience has made the animals bolder than before) (Fig 1, lines 159-161, Fig. S1). I did not notice this possibility mentioned in the discussion of the results.I also do not fully understand this argument. Could you please explain further?

We accept the reviewer's comment and updated the manuscript (lines 336346) explaining the two hypotheses more clearly and arguing that it is difficult to tell them apart with the current data.

[Minor] You also say that "this difference... can be seen in Figure 2 when examining only the bats that had remained until the last trial (Figure 2A2)." Do you mean supplementary Figure S1 A2? In fact, I am entirely unclear on what data is plotted in the supplementary Figure S1 and what differentiates the two columns of figures and the two models presented in the supplementary table. Did you plot data similar to that in Figure 2, with only bats that were present for all trials, but not show this data?

There was a mistake: what was previously referred to as 2A2 is actually S2 A2.

On the right side—only among the individuals with GPS data—the change is already evident at Baseline 2, where only the bolder individuals remain. If you have suggestions for a better analysis approach, we would be happy to hear them.

### Minor pointsGeneral points regarding figures:For Figures 2 and 3A1-3 (as well as Fig. S1): Authors must show the raw data points over the box plots. It is very difficult to interpret the data and conclusions without being able to see the true distribution.

Done

For all figures showing grouped individual data, please annotate all panels or sets of boxplots with the number of bats whose data entered into each, as it is a little difficult to keep track of the changing sample sizes across experimental stages.

To enhance transparency, we have added individual data points to all boxplots, allowing visual estimation of sample sizes across experimental stages. While numerical annotations are not included on the figures, the exact number of bats contributing to each group is provided in the Methods section (Table 8), ensuring this information is readily accessible to readers.In response to the reviewer’s request, we have updated all relevant figures to display individual data points within each boxplot. This addition makes it easier to track changes in sample size across different experimental stages.

Unless I've missed the reason behind differences in axis labelling across the figures, it seems that trials are not always referred to consistently. E.g. Fig. 1 labels say "Trial 1 (baseline)" and fig. 2 labels say "Baseline 1 0 days." I'm not entirely sure if these correspond to exactly the same data. If so, perhaps the labels can be made uniform. I think the descriptive ones (Baseline 1, Postenrichment...) may be more helpful to the reader than providing the trial number (Trial 1, etc....).

Done

Figure 1:Very good Fig. 1A and 1B.For panels C1-3 & D, I think it would make it easier for the reader if the personality measure labels were placed at the top of each panel, e.g. "Boldness (entrance proportion)". The double axis labels are not only harder to read, they are also redundant, as the personality measure label repeats on both axes.

Done

Panel C1: For the first panel in this sequence, I think it would be elegant to include an annotation in the figure that indicates what the datapoints lying on either side of the dashed line means, i.e. "bolder after enrichment treatment" in the upper left corner, and "bolder before enrichment treatment" in the bottom right corner.Panel C2: It appears as though many of the data points in this panel overlap, and it appears to me that the blue data points in particular are overlaid by the orange ones. I am guessing this happens because proportion values based on entrances to only 6 boxes end up giving a more "discrete" looking distribution. I wonder if you can find a way to allow all the data to be visible by, e.g., jittering the data slightly; if there is rounding being done to the proportions, perhaps don't round them so that minute differences will allow them to escape the overlap; or possibly split the panel by enrichment treatment.Caption for C1-3: it may be helpful to mention the correlation line color scheme: "enriched (blue lines), the impoverished (orange lines)". The caption also says positive correlations were found for "both environments together," but this correlation line is not shown. Perhaps mention "(not shown)" or show line. Please rephrase the sentence "Dashed line represents the Y=X line." for more transparency and clarity. I understand you mean an "equality" or "unity" line, but perhaps you can explicitly state the information that this line provides, something like e.g. "Dashed line indicates equal values measured on both trials."

We added the line for a reference, the caption was corrected

Figure 3:Panels B1-C2: I would suggest giving these panels supertitles that indicate that B panels are enriched, C panels are impoverished, and that each panel is data from a different individual.

The legend was corrected to be more clear about the figure

General points regarding tables:Please revisit tables for formatting and typos, particularly in Table 4. Please also revise table captions for clarity. E.g. "first exploration as predisposition" to "Exploration (Baseline 1)" or similar

Done

Supplementary Tables and Figure: these are missing captions and explanations.

The missing parts were adddad and corrected

Points of clarification/style:It would seem to me more logical to present the results shown in Table 3 before those in Table 2, given that the primary in-lab manipulation is discussed with relation to Table 3, and the analysis in Table 2 is discussed rather as a limitation (though I believe this result can be expanded upon further, see above).For the activity metric, I would suggest showing this data as actions/hour instead of actions/minute. I think it is much more intuitive to consider, for example, that a bat makes 2 actions every hour, than that it makes 0.002 actions per minute.

Done